# Evaluation of Heat Stress Effects in Different Geographical Areas on Milk and Rumen Characteristics in Holstein Dairy Cows Using Robot Milking and Rumen Sensors: A Survey in South Korea

**DOI:** 10.3390/ani12182398

**Published:** 2022-09-13

**Authors:** Jang-Hoon Jo, Jalil Ghassemi Nejad, Jae-Sung Lee, Hong-Gu Lee

**Affiliations:** Department of Animal Science, Konkuk University, Seoul 05029, Korea

**Keywords:** smart farming system, dairy cows, rumen sensor, automatic milking, rumen temperature and activity, milk quality

## Abstract

**Simple Summary:**

The purpose of this study was to objectively evaluate the degree of damage to Holstein cows in Korea caused by summer heat stress. It was also established that the milk and rumen characteristics changed under heat stress depending on the difference in the Holstein cows’ parity. As a result of the study, it was confirmed that the summer weather in Korea adversely affects the milk yield, milk fat, milk protein, somatic cells, rumen activity, and rumen temperature of Holstein cows. Additionally, a correlation was found between the degree of heat stress experienced by Holstein cows based on parity. With AMSs and rumen biosensors, this study could provide farms with advice on improving milk yields in Holstein dairy cows. The results of this study suggest that the metabolic mechanisms of each of these factors are needed in Korea to understand how they contribute to the maximum improvements in milk yield and characteristics.

**Abstract:**

This survey investigated, using robotic milking and rumen sensors, the effects of an adjusted temperature–humidity index (THI) in different geographical areas on milk yield, fat and protein, rumen temperature, and activity in lactating Holstein cows. We additionally explored the effect of parity on milk and rumen temperature and activity under different THI levels during the summer. From January to September 2020, four farms (276 dairy cows) were subjected to the use of robot milking machines, and two farms (162 dairy cows) to the use of rumen sensors. For the temperature and humidity data, the THI was calculated on the basis of the data from the Korea Meteorological Administration (KMA). The data were analyzed using the GLM procedure of SAS. Milk yield and milk protein decreased (*p* < 0.05), and milk fat increased (*p* < 0.05) at all farms during the summer, from July to August, when the temperature and humidity were high (THI = 72–79). Milk yields were the highest in the fifth, sixth, seventh, and eighth parities, and the lowest in the fourth (*p* < 0.05). Milk fat concentration was the highest in the fourth parity and the lowest in the first parity (*p* < 0.05). In the first parity, the highest levels of milk protein and lactose were seen (5.24% and 4.90%, respectively). However, milk protein concentration was the lowest in the third parity, and the lactose concentration was the lowest in the fifth, sixth, seventh, and eighth parities. According to the rumen sensor, the rumen temperature of the dairy cows at the two farms also continued to increase (*p* < 0.05) from July to August, and then decreased (*p* < 0.05) in September. However, the activity in the rumen was increased (*p* < 0.05) from July to September. In the second parity, the highest rumen temperature (39.02 °C) was observed, while the lowest value (38.28 °C) was observed in the third parity. The highest value of rumen activity (12.26 mg) was observed in the second parity and the lowest value (11.31 mg) in the fourth parity. These data, taken together, confirm that a high THI during summer conditions negatively affects milk yield, milk protein content, and rumen temperature and activity in lactating Holstein cows. It is also demonstrated that various parities affect milk characteristics and the rumen environment in the summer season.

## 1. Introduction

Smart farming (e.g., using robotic milking for animal welfare and production management, and using a rumen sensor to monitor rumen activity during seasonal changes and when feeding cows different rations) has the potential to reduce the impact of the environment, such as workers, ration, and seasons, on livestock production quality [1]. Many farms prefer to use robotic milking and rumen sensors for two major reasons: (1) to improve animal welfare and (2) to increase the farm’s economic efficiency. It is worth noting that automatic milking systems (AMSs) are not only an alternative to traditional milking methods, but also a broader approach to managing dairy herd health, welfare, and productivity [1]. According to Wade et al. [2], milk yield was increased by 12.4% instead of 2% when an AMS was implemented, without accounting for the year effect.

Biosensors are becoming increasingly important in the livestock sector [3]. They can be used to measure physiological, behavioral, immunological, and other variables in animals. Compared to other methods of measuring rumen characteristics, rumen sensors offer several advantages. It is particularly important to monitor rumen kinetics continuously and at a high resolution using rumen sensors. These sensors generate an enormous amount of data, which need to be properly processed and interpreted to cull out critical information on the metabolic status of the rumen and the host animal.

Optimizing the rumen environment can help improve milk production and milk characteristics, both of which are considered critical to achieving farm profitability, and are highly desired by the vast majority of farmers who use automatic milking systems and rumen sensors [3,4]. However, there are also numerous negative effects of environmentally induced hyperthermia on the dairy industry, such as economic and productivity loss, and a negative influence on animal welfare [5]. Heat stress (HS) reduces the energy intake of lactating dairy cows, resulting in their inability to meet their bodies’ demands for maintaining milk production and health. Due to this condition, milk yields and quality are decreased, and the animals are more prone to diseases. [5]. While most studies on AMSs have been conducted in Europe, there is a relative lack of studies on AMSs in South Korea, where the summers are hot and humid. In contrast to other regions, South Korea is surrounded by water, thus the temperature and humidity are high. Given the aforementioned information, South Korea has different environmental circumstances compared to other countries, making the country’s temperature and humidity essential while researching heat stress in Holstein cows. The critical temperature–humidity index (THI) for animal well-being might be exceeded during the summer months. A thermoregulatory behavior has been observed in cows, as well as in other domesticated ruminants, in the hotter hours of the day [6], which can negatively impact their well-being, and consequently their performance. The response of an animal to heat stress is also influenced by parity [7]. Milk and rumen parameters have been poorly studied in spite of the growing popularity of AMSs and rumen sensors. Additionally, there are not many studies that discuss the use of rumen sensors and robot milking systems when dairy cows are subjected to heat stress (HS) in the summer. 

Therefore, the objectives of this study were (1) to investigate, using robotic milking and rumen sensors, the influence of an adjusted THI in different geographical areas on milk yield, milk fat and protein, somatic cells, rumen temperature, and rumen activity in lactating Holstein cows, and (2) to compare the milk and rumen performance of dairy cows triggered by parity numbers during summer in Korea.

## 2. Materials and Methods

### 2.1. Experimental Design and Animals

This survey did not require the approval of the local ethical committee, in accordance with applicable law. It was routine for the herd’s activities to follow all of the procedures included in this survey. We studied 276 Holstein Friesian cows that were subjected to the use of robot milking machines, and 162 Holstein Friesian cows that were subjected to the use of rumen sensors in tie stall barns located in different geographical areas in Korea. Animal breeding environments were met (feed and water, and the regular health check), and total mixed ration (TMR) was fed. All farms had identical TMR compositions, which included forage to concentrate ratios of 60:40, and 14–15% crude protein in diets. An additional concentrated mixture was offered to cows individually in the milking box according to their daily milk yield. We chose farms with comparable housing conditions, so the housing system was the same. From January to September 2020, four farms (two farms in Pocheon (Farms A and B), one in Pyeongtak (Farm C), and one in Hwaseong (Farm D); a total of 276 dairy cows) were subjected to the Lely-Astronaut A4 automatic milking system (AMS; Maassluis, Netherlands), and two farms (Pocheon, a total of 162 dairy cows) were subjected to a sensor (Smaxtec, Graz, Austria) inserted in the rumen. The cows had free access to fresh water at all times.

### 2.2. Sample and Data Collection

Daily milk yield and milk composition were measured by the robot milking system twice a day. The rumen biosensor was implanted by qualified veterinarians through the mouth, and data on rumen activity and temperature were collected using a software program (Smaxtec, Austria). Temperature and humidity were set based on the Korea Meteorological Administration (Table 1). Two data loggers (WST 1800; MTX Italia srl, Modena, Italy) in the middle of the barn, positioned between each resting and feeding area, recorded the temperature and relative humidity (RH), and the data were recorded hourly during the study. Daily average environmental data from all sides of the barn were not significantly different; thus, the means were used. The following equation, proposed by Jo et al. [5], was used to calculate the temperature–humidity index (THI):THI = (1.8 × T_db_ + 32) − (0.55 − 0.0055 × RH/100) × [(1.8 × T_db_ − 26)],(1)
where T_db_ is the dry bulb temperature (°C) and RH is the relative humidity (%). We subjected the dairy cow to THI conditions, which change based on temperature and humidity, including Stress Threshold: 68~71 THI, Mild–Moderate Stress: 72~79, Moderate–Severe Stress: 80~89 THI, Severe Stress: 90~98. In response to the refusal of visits to the robot, basic descriptive statistics were provided for both milk yield and composition. In the original data set, 276 Holstein cows were observed from January to September 2020. The milk yield, milk protein and fat, and somatic cell counts (SCCs) were measured each day by adding up all the milk collected over two milking activities a day (a 24 h period). The milk composition determined from the AMS was calculated using the 24 h average of milk yield at the time of the corresponding milking. On the days on which milk yield was recorded, the AMS software automatically retrieved information on milking frequency by counting the number of events for the previous 24 h. Using the starting and end times of the milking event, the length of time between two consecutive milkings was computed, and the daily average interval between milking was obtained. Because just a few parameters, such as fat, protein, and the SCC in the available data set were monitored, we discuss a limited number of milk characteristics. The 3.5% fat-corrected milk (FCM) and energy-corrected milk (ECM) yields were calculated using the following equation: 3.5% FCM: (0.4324 × milk yield) + (16.216 × milk fat yield), ECM: (0.327 × milk yield) + (12.95 × milk fat yield) + (7.2 × milk protein yield). With regard to the rumen temperature and activity, we recorded the sum of consecutive measurements taken over a 24 h period as the sum of the daily rumen temperature and activity, as suggested by Ramunas et al. [8]. Briefly, the rumen sensors to measure rumen temperature and activity were implanted in reticulo-rumen regions of lactating Holstein cows in bolus form (TX-1442A, Smaxtec Animal Care, Austria). As per manufacturer recommendations, a standard balling gun was used to implant the bolus via the oral route while the dairy cow was restrained in a squeeze chute. The size and ruminal fluid resistance of the bolus device were approximately 105 mm × 35 mm (length × diameter) and 0.2 kg, respectively. The temperature sensor could detect temperatures between 0 °C and 50 °C (±0.05 °C). By using accelerometers (located inside the bolus) to measure animal movement, an activity index (between 0 and 100 percent) was calculated. The temperature and activity of the reticulo-rumen were measured every 10 min. The antenna operated in a 30 m range. Smaxtec Messenger (Smaxtec Messenger, Smaxtec Animal Care, Austria) provides access to the data collected by the antenna via a radio signal and online servers (with cloud storage). Each parameter was analyzed independently of the following factors: The number of milking events in subsequent parity were 1 (*n* = 77), 2 (*n* = 70), 3 (*n* = 64), 4 (*n* = 29), and ≥5 (*n* = 35).

### 2.3. Statistical Analysis

The GLM procedure of SAS was used to conduct the statistical analysis (Studio Version, SAS Institute Inc., Cary, NC, USA). The analysis incorporated repeated measurements of properties such as milk yield, milk price, milk protein, milk fat, somatic cell, rumen temperature, and activity; and animal distribution into treatment was regarded as a random effect. The covariance structure of each variable was examined using four elements (compound symmetry, autoregressive order 1, unstructured covariance, and variance components). The mean value for each trait was compared using the covariance structure with the lowest Akaike information criterion. The difference between the means of the data was considered statistically significant when the *p*-value was less than 0.05, and to have a significant trend tendency when the values were between 0.05 and 0.10. The standard error of the mean is reported. The correlation coefficients for the milk characteristics and rumen environment were calculated on the basis of Pearson’s correlation coefficients.

## 3. Results

### 3.1. Milk Yield and Characteristics

The study shows that in the summer, when the temperature and humidity were high, from July to September, the milk yield decreased (*p* < 0.05; Table 2) in all farms (11%). At the same time, the milk protein decreased (*p* < 0.05; Table 3) and the milk fat (*p* < 0.05; Table 3) and the SCC (*p* < 0.05; Table 3) increased in all farms.

According to parity number, Table 4 presents the mean values and standard deviations of the analyzed traits. The parity numbers 5, 6, 7, and 8 were grouped together and were considered as a single set. The impact of the parity number on all studied attributes was determined to have statistical significance. The highest milk yield (36.11 kg) was observed in the fifth, sixth, seventh, and eighth parities, and the lowest (30.13 kg) in the fourth parity. The highest somatic cell count was shown to be in the fifth, sixth, seventh, and eighth parities (224.16 × 10^3^/mL).

The levels of milk fat were the highest (4.28%) in the fourth parity and the lowest (3.76%) in the first parity. The milk protein and lactose levels were the highest (5.24% and 4.90%, respectively) in the first parity. However, milk protein was the lowest (3.05%) in the third parity, and lactose was the lowest (4.77%) in the fifth, sixth, seventh, and eighth parities.

There is a correlation matrix in Tables 7 and 8 for the traits considered for milking. Statistical significance was determined for all correlation coefficients (*p* < 0.05). At Farm A, milk yield was strongly and positively correlated with MPY, MFY, 3.5% FCM, and ECM (0.989, 0.986, 0.996, and 0.997, respectively), and moderately correlated with milk protein (0.345). However, MPY and milk protein showed a positive correlation (0.510 and 0.705, respectively), and MFY, 3.5% FCM, and ECM showed a negative correlation (−0.408, −0.194, and −0.029, respectively). At Farms A and B, milk yield was moderately and negatively correlated with milk fat (−0.868 and −0.603, respectively). There was a strong positive correlation between milk yield, milk protein, and MPY.

### 3.2. Rumen Temperature and Activity

At the two farms where the rumen sensors were being used, the rumen temperatures continued to increase (*p* < 0.05) from July to August, and then decreased (*p* < 0.05) in September (Table 5). The activity in the rumen increased (*p* < 0.05) from July to September (Table 5).

Table 6 shows the mean values and standard deviations of the qualities studied in relation to the parity number. The parity numbers 5, 6, 7, and 8 were grouped together and considered as a single set. The impact of the parity number on all studied attributes was determined to have statistical significance. The highest rumen temperature (39.02 °C) was observed for the second parity, and the lowest (38.28 °C) for the third parity. Rumen activity was the highest (12.26 mg) in the second parity and the lowest (11.31 mg) in the fourth parity.

As shown in Table 7 and Table 8, we calculated the correlation matrix of all milking traits. Milk yield showed a negative correlation with rumen temperature in cows at Farm A (−0.473), and a positive correlation in cows at Farm B (0.267). Rumen activity showed a negative correlation at Farms A and B (−0.033 and −0.424, respectively).

## 4. Discussion

### 4.1. Milk Yield and Characteristics

According to the cluster patterns observed over the last 20 years, research has tended to focus on animal-related issues and process implementation [1]. Several considerations associated with animal clusters in terms of welfare and behavior have appeared at a high frequency (3.2% and 4.1%, respectively) [1]. In some instances, observing cows for their health and welfare less frequently could result in a further decline in milk quality, which could worsen the situation. The AMS depends on a qualified service being available at all times [9]. Technological proficiency and accustoming the animals to the equipment should also be considered. According to our findings, the AMS can boost milk supply though parity, and environmental factors can influence this phenomenon. In line with our findings, Richard et al. [10] showed that while using AMS, milk yield decreased when the THI was 83 or higher, or when it was 72 or lower. When the THI exceeds 70~72, cows tend to experience HS [11]. A correlation was found between THI and milk yield [12]. During summers, due to high temperatures and humidity, farmers suffer economic losses. Environmental parameters (THI) and milk characteristics exhibit a distinct variation during the seasons, which is corroborated in this study. The summer of 2020 in Korea was associated with higher HS risks for cows. To cope with HS, cows generally consume more feed at night than during the day. However, if the temperature is high during the night, they are unable to do so. Given this, it is important to develop cost-effective methodologies to minimize HS effects during high THI times [13]. It is also important to monitor parity because it allows for a better control of milk output and AMS performance. The estimation of parity curves using conventional milking systems has been tested, but prediction models for AMS are still being developed [14]. Parity number has been studied extensively in relation to dairy cattle milk yield during both gestation and parturition. Studies have demonstrated that the conventional milking method maximizes milk yield within the fourth or fifth parity when parity numbers increase [15]. In our study, milk yield increased with increasing parity number, presumably due to growing udder size and development, as well as an increase in secretory cells [16]. Increasing parity between primiparous and multiparous cows can also contribute to a high milk yield by influencing tissue mobilization and increasing the body weight of dairy cows over the first parity [17]. Our results reveal that first-parity cows have lower milk yields due to a lack of development of the mammary gland and the mammary vein at this stage, which could be explained by the fact that the cows are not in the productive stage at the time.

Due to seasonal variables, the quality of milk produced by dairy cows might fluctuate over time [5]. For the dairy cow industry, nutrients, health, and milk cost/income are critical and can be influenced by a variety of factors, including management, health, lactation stage, and environment [18,19]. It appears that the most important factor is the variation in milk yield and compositions with respect to milking time [20]. Rajcevic et al. [21] found that milk protein levels are the highest in winter and the lowest in summer. They suggested that the correlation between protein and casein contents is the strongest. According to Bernabucci et al. [22], summer milk contains lower fat and protein concentrations than the milk in other seasons. Fat and protein concentrations reportedly decrease as the temperature increases [23]. Among the farms in the Lombardy region (in Italy), Bernabucci et al. [24] found that the THI was negatively correlated with the fats and proteins in milk, with breakpoint at 50.2 and 65.2 maximum THIs, respectively. Additionally, in a study conducted on Georgian and Israeli primiparous and multiparous dairy cows, Aharoni et al. [25] observed an increase in fat and protein concentrations from October to January, followed by a fall in spring and a significant reduction in summer. However, in our study, milk fat increased in summer, which may have been caused by long-term HS, leading to the decomposition of long-chain fatty acids, and thus, increased milk fat. Furthermore, the higher percentage of fat in the milk of heat stressed cows could be attributed to the reduction in milk yield and subsequent concentration of fat, in addition to possibly greater non-protein nitrogen contents in the milk produced from cows under HS [26]. Several studies have shown that milk protein and lactose concentrations are higher during the first parity than the third parity [24,27]. In addition, there is a general decrease in fat and protein contents during the summer, which could be linked primarily to the hot weather negatively influencing the synthesis of these components. When milk protein is taken into account, first-parity cows seem more vulnerable to HS. However, it is hard to explain the opposite response. The first-parity cow should be able to sustain its growth [17]. HS may also have contributed to the drop in milk yield in these animals, as lactose is the primary osmotic regulator of milk volume [28]. In addition, severe HS negatively impacted goat milk lactose concentration and mammary glucose uptake [29]. HS has a detrimental influence on the daily SCC in dairy cows in Mediterranean climates, according to Bouraoui et al. [30], with a higher SCC in the summer. The SCC decreased as the THI decreased in September, but the results may vary, depending on the management and environment in each farm. In our study, under HS, the SCC in the third parity was higher than that in the first and second parities, which is consistent with the results of some previous works [31,32]. In a study on Holstein cows, consistent with our result, under HS, the SCC was higher in the third parity than in the first and second parities [7]. The correlation between milk yield and milk composition found in this study showed high positive values. Similar to our study, other studies have shown a high correlation between milk yield and milk compositions [33,34].

### 4.2. Rumen Temperature and Activity

In the livestock sector, it is becoming increasingly important to use rumen biosensors. Rumen sensors can measure a variety of variables, including physiological, immunological, and behavioral data [3]. The use of rumen sensors over alternative measures of rumen characteristics provides several advantages. Rumen sensors are especially advantageous for monitoring pH and other ruminal kinetic characteristics in a continuous, high-resolution manner [3]. A reticulum is usually where sensors reside in non-cannulated animals [3]. According to Gonzalez-Rivas et al. [35], there is a high positive correlation between rumen temperature and THI. Similar to our study, Liang et al. [36] observed that the effect of climate conditions on the rumen temperature was higher in summer (40.4 °C) than in spring and fall (40.1 °C), or winter (40.0 °C). As a result of the heat generated during the fermentation process, the temperature in the rumen of the dairy cows is generally higher than that in the other body regions [37]. In summer, the average daily rumen temperature increased gradually when the THI was classified as lower than 60, 60 to 65 (39.2 °C), 65 to 70 (39.3 °C), and higher than 70 (39.4 °C). In addition, the average temperature per daily volume of rumen was higher in summer than in spring and winter. Rumen temperatures are usually about 0.5 °C greater than body core temperatures, due to the presence of heat-producing microorganisms in the rumen [8]. In our study, the rumen temperature was higher in the second parity than in the first parity in summer. The rumen temperature of multiparous cows was shown to be higher than that of primiparous cows in early lactation Holstein cows [38]. Humer et al. [38] stated that parity affects reticuloruminal temperature. In their study, they found that older cows spent 1 h more per day with a reticuloruminal temperature >39.5 °C, when compared with primiparous animals. We also hypothesized that the microbial activity of the reticulorumen might be higher because multiparous cows produce at a rate that is higher than that of primiparous animals. Multiparous cows appear to produce more heat through fermentation, as a result of their increased feed intake. In line with our study, Bewley et al. [37] also showed a negative correlation between rumen temperature and milk yield. The milk yield of cows is associated with increased metabolic heat, and cows with high milk yields are more sensitive to HS, so animal performance may be a major factor in determining the temperature in the rumen [36]. The temperature in the rumen continues to rise due to HS, and there are changes in the feed intake and the microbial compositions in the body to reduce heat generation.

The rumen activity in ruminant animals can be a reliable early indicator of certain metabolic diseases, such as ruminant acidosis [39]. Recently, Stangaferro et al. [40] monitored rumen activity to identify cattle with health problems. In the future, to help with health management decisions, sensors are likely to be used to monitor the amount of rumen activity, taking into account physiological factors (e.g., rumen temperature). According to our findings, when the THI was higher in July than in September, the rumen activity was lower. Consistently, rumen activity was shown to be lower when the THI was 79~83 than when the THI was 58.8~66.5 [41]. Other investigations have found decreased rumen activity in animals under HS, indicating a negative relationship between THI and rumen activity [42]. Rumen activity can be affected by several environmental factors, such as nature, milking time, and photoperiod [43,44]. Calving prediction should take into account rumen temperature and activity [45]. A precision technique that assists in automated detection facilitates herd management by accurately predicting events associated with animal reproduction, especially calving. In the following stages, further research into dairy cow reproduction with the aid of a rumen sensor is needed. HS reduces blood flow to the rumen epithelium and inhibits reticular motility and rumination. HS also decreases feed intake, and the activity of microorganisms in the rumen decreases due to an imbalance in the nutrients in the body, thereby decreasing rumen activity [41]. Additionally, HS causes changes in rumination activity and motility, as well as microbiota, as a result of feed digestion and rumination fermentation [46]. An increase in the environmental temperature directly affects the appetite center of the hypothalamus [47], reducing rumination time [42] and suppressing appetite [48]. A decrease in rumen activity can also affect the gastrointestinal digestion rate [49]. In addition, the rumen produces less volatile fatty acids when the ruminal activity is decreased [50]. This area is complex, due to the high number of biologically active compounds produced in the rumen. Taken together, a higher HS has an impact on rumen activity and temperature in dairy cows. Therefore, to maintain homeostasis, it is important to reduce heat production in the rumen, with optimum microbial activity to minimize heat dissipation in dairy cows.

## 5. Conclusions

The findings of this study provide insights into the effects of high THI during summer conditions on milk yield, milk fat and protein contents, rumen temperature, and rumen activity in lactating Holstein cows. Using a robot milking system and rumen sensor, it was possible to confirm monthly changes in milk compositions and rumen function due to HS. It was also demonstrated that milk characteristics and the rumen environment are different under the same HS, depending on the difference in the parity of dairy cows. Although rumen temperature changed with monthly changes in the THI, continual HS in July and August may have reduced milk yield and protein in September, decreasing income from milk for farmers. We suggest that robot milking systems and rumen sensors be used to improve our understanding of summer HS effects on lactating cows. This study could be used as a benchmark to advise farms on how high milk yields can be achieved in Holstein dairy cows using AMSs and rumen biosensors. It is important to further investigate the molecular mechanisms of each of these factors and how they contribute to the maximum milk yield in Holstein dairy cows in Korea, and possibly in other parts of the world.

## Figures and Tables

**Table 1 animals-12-02398-t001:** Observed mean monthly temperature–humidity index (THI) based on the data from the year 2020 of the Korea Meteorological Administration.

THI	Jan	Feb	Mar	Apr	May	Jun	Jul	Aug	Sep
Regions									
Pocheon									
Temperature (°C)	−1.19	0.01	5.50	9.14	16.89	22.51	22.75	25.10	18.77
Humidity (%)	79.30	78.73	62.58	57.69	79.15	79.32	88.92	96.01	87.54
THI	33.08	35.29	45.07	50.80	61.81	70.76	72.02	76.72	65.35
Pyeongtaek and Hwaseong									
Temperature (°C)	1.59	2.39	7.08	10.50	17.61	23.04	23.80	26.75	21.08
Humidity (%)	77.72	81.04	71.21	69.33	78.75	77.01	82.77	88.40	78.51
THI	37.74	38.81	46.79	52.17	62.90	71.44	73.19	78.66	68.57

THI = (1.8 × T_db_ + 32) − [(0.55 − 0.0055 × RH) × (1.8 × T_db_ − 26)] [5]; Jan, January; Feb, February; Mar, March; Apr, April; Jun, June; Jul, July; Aug, August; Sep, September.

**Table 2 animals-12-02398-t002:** Milk yield of lactating Holstein cows using the robot milking system at different farms from January to September.

	Jan	Feb	Mar	Apr	May	Jun	Jul	Aug	Sep	SEM ^1^	*p*-Value
Milk yield (kg)											
Farm A	36.52 ^a^	36.93 ^a^	37.18 ^a^	36.66 ^a^	34.56 ^b^	34.57 ^b^	34.23 ^b^	30.31 ^c^	28.85 ^d^	0.159	<0.001
Farm B	37.66 ^a^	36.99 ^a^	37.12 ^a^	37.76 ^a^	38.47 ^a^	38.04 ^a^	36.55 ^a^	32.68 ^b^	31.11 ^b^	0.220	<0.001
Farm C	34.24 ^c^	37.07 ^a^	36.43 ^ab^	36.20 ^ab^	35.92 ^b^	36.33 ^ab^	34.66 ^c^	31.71 ^d^	30.79 ^d^	0.125	<0.001
Farm D	32.37 ^bc^	31.96 ^c^	32.77 ^bc^	32.38 ^bc^	33.17 ^ab^	33.98 ^a^	33.44 ^ab^	28.41 ^d^	29.26 ^d^	0.132	<0.001

Means with different superscripts (a, b, c, and d) differ significantly according to Tukey’s test (*p* < 0.05); ^1^ SEM, standard error mean; Jan, January; Feb, February; Mar, March; Apr, April; Jun, June; Jul, July; Aug, August; Sep, September; Farm A, Gyeong-gi-do Pocheon; Farm B, Gyeonggi-do Pocheon; Farm C, Gyeonggi-do Pyeongtaek; Farm D, Gyeonggi-do Hwaseong.

**Table 3 animals-12-02398-t003:** Milk characteristics of lactating Holstein cows using the robot milking system at different farms from January to September.

	Jan	Feb	Mar	Apr	May	Jun	Jul	Aug	Sep	SEM ^1^	*p*-Value
Milk protein (%)											
Farm A	3.27 ^a^	3.30 ^a^	3.27 ^a^	3.22 ^b^	3.14 ^d^	3.18 ^c^	3.18 ^c^	3.16 ^cd^	3.09 ^e^	0.004	<0.001
Farm B	3.19 ^bcde^	3.20 ^abcd^	3.21 ^abc^	3.22 ^ab^	3.23 ^a^	3.18 ^cde^	3.17 ^de^	3.16 ^e^	3.04 ^f^	0.004	<0.001
Farm C	3.25 ^a^	3.24 ^abcd^	3.09 ^b^	3.03 ^c^	3.02 ^c^	2.95 ^d^	2.96 ^d^	2.96 ^d^	2.95 ^d^	0.008	<0.001
Milk fat (%)											
Farm A	4.18 ^abc^	4.19 ^ab^	4.15 ^bc^	4.17 ^abc^	4.18 ^abc^	4.13 ^c^	4.07 ^d^	4.20 ^ab^	4.22 ^a^	0.004	<0.001
Farm B	3.79 ^c^	3.97 ^bc^	4.03 ^bc^	4.06 ^b^	3.98 ^bc^	3.77 ^c^	3.85 ^bc^	4.34 ^ab^	4.09 ^ab^	0.022	<0.001
Farm C	4.13 ^cd^	3.64 ^e^	4.15 ^cd^	4.41 ^bc^	4.46 ^b^	3.91 ^de^	3.97 ^d^	4.05 ^d^	5.24 ^a^	0.026	<0.001
Somatic cell (1000/mL)											
Farm A	88.66 ^d^	138.55 ^bc^	119.22 ^cd^	159.05 ^ab^	156.81 ^ab^	186.42 ^a^	166.26 ^ab^	157.18 ^ab^	93.85 ^d^	2.756	<0.001
Farm B	160.79 ^b^	171.26 ^b^	177.03 ^b^	132.91 ^b^	125.73 ^b^	130.53 ^b^	97.24 ^b^	135.05 ^b^	308.70 ^a^	9.637	<0.001
Farm C	201.13 ^a^	138.84 ^b^	140.91 ^b^	122.37 ^bc^	119.58 ^bc^	103.75 ^cd^	126.70 ^bc^	122.59 ^bc^	74.44 ^d^	2.710	<0.001

Means with different superscripts (a, b, c, d, e and f) differ significantly according to Tukey’s test (*p* < 0.05); ^1^ SEM, standard error mean; Jan, January; Feb, February; Mar, March; Apr, April; Jun, June; Jul, July; Aug, August; Sep, September; Farm A, Gyeonggi-do Pocheon; Farm B, Gyeonggi-do Pocheon; Farm C, Gyeonggi-do Pyeongtaek.

**Table 4 animals-12-02398-t004:** Mean and standard deviations (SDs) of considered traits for milking in the groups designated by parity number in the South Korean summer.

Trait					
	Milk Yield (kg)	Somatic Cell	Milk Fat	Milk Protein	Lactose
1st parity					
Mean	30.68 ^c^	85.24 ^c^	3.76 ^b^	5.24 ^a^	4.90 ^a^
SD	1.451	30.952	0.122	0.868	0.021
2nd parity					
Mean	34.13 ^b^	88.44 ^c^	4.27 ^a^	3.08 ^b^	4.84 ^b^
SD	2.488	18.266	0.542	0.025	0.03
3rd parity					
Mean	31.35 ^c^	167.19 ^b^	4.15 ^a^	3.05 ^b^	4.80 ^c^
SD	3.232	39.143	0.178	0.028	0.042
4th parity					
Mean	30.13 ^c^	187.95 ^b^	4.28 ^a^	3.1 ^b^	4.78 ^cd^
SD	3.608	51.183	0.126	0.032	0.063
5th, 6th, 7th, and 8th parities					
Mean	36.11 ^a^	224.16 ^a^	3.87 ^b^	3.1 ^b^	4.77 ^d^
SD	3.602	118.493	0.253	0.052	0.062

Mean values differing statistically significantly between teats are marked by different letters (a, b, c, and d); *p*-value < 0.05. N = number of daily milking events. Number of milking events in subsequent parity: 1 (*n* = 77), 2 (*n* = 70), 3 (*n* = 64), 4 (*n* = 29), and ≥5 (*n* = 35).

**Table 5 animals-12-02398-t005:** Rumen temperature and activity of lactating Holstein cows using rumen sensors at different farms from July to September.

	Jul	Aug	Sep	SEM ^1^	*p*-Value
Rumen temperature (°C)					
Farm A	38.92 ^b^	39.00 ^a^	38.89 ^c^	0.007	<0.001
Farm B	36.71 ^b^	37.20 ^a^	36.13 ^b^	0.057	<0.001
Rumen activity					
Farm A	11.84 ^b^	12.09 ^ab^	12.21 ^a^	0.056	0.042
Farm B	11.63 ^b^	11.64 ^ab^	11.80 ^a^	0.032	0.016

Means with different superscripts (a, b, and c) differ significantly according to Tukey’s test (*p* < 0.05); ^1^ SEM, standard error mean; Jul, July; Aug, August; Sep, September; Farm A, Gyeonggi-do Pocheon; Farm B, Gyeonggi-do Pocheon.

**Table 6 animals-12-02398-t006:** Mean and standard deviations (SDs) of considered traits for rumen in the groups designated by parity number in the South Korean summer.

	Trait	
	Rumen Temperature	Rumen Activity
1st parity		
Mean	38.95 ^b^	11.81 ^b^
SD	0.108	0.486
2nd parity		
Mean	39.02 ^a^	12.26 ^a^
SD	0.109	0.903
3rd parity		
Mean	38.28 ^c^	11.51 ^c^
SD	0.294	0.845
4th parity		
Mean	38.96 ^ab^	11.37 ^ab^
SD	0.113	0.627
5th, 6th, 7th, and 8th parities		
Mean	38.93 ^b^	12.13 ^b^
SD	0.107	0.779

Mean values differing statistically significantly between teats were marked by different letters (a, b and c); *p*-value < 0.05. N = number of daily milking events.

**Table 7 animals-12-02398-t007:** Pearson’s correlation coefficient of considered traits for daily milking and rumen characteristics at Farm A.

	Milk Yield	Somatic Cell	Milk Fat	Milk Protein	MPY	MFY	3.5% FCM	ECM	Rumen Temperature	Rumen Activity
Milk yield	1	0.301	−0.868	0.345	0.989	0.986	0.996	0.997	−0.473	−0.033
Somatic cell		1	−0.143	0.551	0.364	0.341	0.323	0.334	−0.250	−0.013
Milk fat			1	−0.203	−0.844	−0.775	−0.820	−0.827	0.408	0.141
Milk protein				1	0.471	0.367	0.359	0.388	−0.307	−0.193
MPY					1	0.980	0.988	0.993	−0.493	−0.052
MFY						1	0.997	0.995	−0.464	−0.004
3.5% FCM							1	0.999	−0.468	−0.016
ECM								1	−0.474	−0.026
Rumen temperature									1	0.140
Rumen activity										1

Correlation coefficients marked in red are not statistically significant. MPY, milk protein yield; MFY, milk fat yield; FCM, fat-corrected milk; ECM, energy-corrected milk.

**Table 8 animals-12-02398-t008:** Pearson’s correlation coefficients of considered traits for daily milking and rumen characteristics at Farm B.

	Milk Yield	Somatic Cell	Milk Fat	Milk Protein	MPY	MFY	3.5% FCM	ECM	Rumen Temperature	Rumen Activity
Milk yield	1	−0.492	−0.603	0.705	0.510	−0.408	−0.149	−0.029	0.267	−0.424
Somatic cell		1	0.146	−0.347	−0.233	−0.158	−0.208	−0.226	−0.291	0.154
Milk fat			1	−0.222	−0.315	0.795	0.537	0.403	0.265	0.137
Milk protein				1	0.344	0.186	0.248	0.282	0.705	−0.610
MPY					1	0.322	0.627	0.738	−0.338	0.017
MFY						1	0.939	0.876	0.394	0.079
3.5% FCM							1	0.988	0.194	0.073
ECM								1	0.102	0.067
Rumen temperature									1	−0.293
Rumen activity										1

Correlation coefficients marked in red are not statistically significant. MPY, milk protein yield; MFY, milk fat yield; FCM, fat-corrected milk; ECM, energy-corrected milk.

## Data Availability

Not applicable.

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
