# Peer review of "Evaluation of Heat Stress Effects in Different Geographical Areas on Milk and Rumen Characteristics in Holstein Dairy Cows Using Robot Milking and Rumen Sensors: A Survey in South Korea"

_animals, 2022, doi:10.3390/ani12182398_

Round 1
Reviewer 1 Report
Introduction
The introduction focuses too much on smart farming and rumen sensors. However, these are only the means of research, not the goal. There is a lack of description of the effect of heat stress on the organism and performance of European-type cattle, especially in hot and very humid climates. The set goals are too broad, which is reflected in further processing.
Materials and methods:
· A more detailed description of the farms is missing (number of animals from individual farms, TMR composition on the farms, housing technology, milk yield etc.) and climatic data of the areas (air temperature and relative air humidity) where the farms are located. Only THI is not enough (Tab 1). It is not clear how air temperature and air humidity contribute to the value.
· How were the conditions of welfare (l.95) fulfilled, what do the authors mean by this?
· The indication of the individual heat stress zone with values for better orientation is missing
Results:
In general, the results are confusing due to setting many goals and dependencies.
· Why is the price of milk listed? This is a completely irrelevant figure considering the objectives of the work. Not rated further.
· Table 4 - it is necessary to state again the number of dairy cows in individual parities
·
Discussion:
· l.247 – 254: This is common knowledge, not new knowledge.
· The increase in milk fat during periods of stress has a very unlikely explanation. Corroborate with scientific publications.
· Why are the findings on goat milk presented? There are plenty of literary sources in the field of cow's milk.
· L.306 – 308 - his statement is not true, literary sources exist
Abstracts:
· L.18 – 19: no molecular mechanisms were pursued.
· L.41 – 45: the conclusion is wrong - the negative impact on milk fat was not confirmed, unlike other studies.
Although the authors obtained very valuable data, the manuscript processing needs to be revised and simpler goals should be set. Manuscripts can be on two topics due to the data obtained.
Author Response
Comments and Suggestions for Authors
Introduction
The introduction focuses too much on smart farming and rumen sensors. However, these are only the means of research, not the goal. There is a lack of description of the effect of heat stress on the organism and performance of European-type cattle, especially in hot and very humid climates. The set goals are too broad, which is reflected in further processing.
Response: Thank you very much for pointing this out. We added information regarding performance changes in dairy cows when receiving heat stress (Line 71-74). Please see the revision file for your approval.
Materials and methods:
- A more detailed description of the farms is missing (number of animals from individual farms, TMR composition on the farms, housing technology, milk yield etc.) and climatic data of the areas (air temperature and relative air humidity) where the farms are located. Only THI is not enough (Tab 1). It is not clear how air temperature and air humidity contribute to the value.
Response: We understand that reviewer was confused. We added temperature and humidity in table 1. Please see the revision file for your approval (Table 1). In addition, we now added clarification sentences into the revised manuscript regarding the TMR compositions and housing system in the farms. Since the farms were in different areas and the exact compositions of TMRs are private information belonging to each farm, we don’t have detailed information. However, we asked farmers to provide us with some data to clarify the general conditions of the housing system and also the TMR compositions. We understand that the TMR compositions including forage to concentrate ratio in all farms were similar (F:C ratio of 60:40), with CP:14~15 (Line 103-106). The housing system were the same as we have chosen the farms that had similar conditions were applied in regard with housing (free-stall housing with open barn having ceiling to use for shade or protecting from rains) (Line 106-108).
- How were the conditions of welfare (l.95) fulfilled, what do the authors mean by this?
Response: Yes, we agree that the reviewer should have been confused. We clarified this in the text (Line 102). Please see the revision file for your approval.
- The indication of the individual heat stress zone with values for better orientation is missing
Response: Thank you for your comment. At Arizona University in the US, we applied the THI conditions that vary depending on temperature and humidity to the dairy cow. Stress Threshold: 68~71 THI, Mild-Moderate Stress: 72~79, Moderate-Severe Stress: 80~89 THI, Severe: Stress: 90~98 (Line 124-127). We also presented this information it in the picture below.
Source: University of Arizona revised heat stress scale (2011)
<Each temperature/humidity couple corresponds a level of thermal stress for the dairy cow.> |
|
Results:
In general, the results are confusing due to setting many goals and dependencies.
- Why is the price of milk listed? This is a completely irrelevant figure considering the objectives of the work. Not rated further.
Response: Yes, we agree with the reviewer. As the reviewer said, we deleted the milk price table and sentences. Please see the revision file for your approval.
Table 4 - it is necessary to state again the number of dairy cows in individual parities
Response: Yes. We added the number of dairy cows by parity in table 4. Please see the revision file for your approval.
Discussion:
- l.247 – 254: This is common knowledge, not new knowledge.
Response: Yes, we have revised the sentence in agreement with the reviewer. We can think of it as verification through a smart sensor that can be evaluated objectively. Please see the revision file for your approval (Line 259-266).
- The increase in milk fat during periods of stress has a very unlikely explanation. Corroborate with scientific publications.
Response: Studies have shown that when stressed, long-chain fatty acids in the body decomposed and milk fat increased. In addition, the higher percentage of fat in the milk of heat stressed cows could be attributed to the reduction in milk yield and subsequent concentration of fat in addition to possibly greater non-protein nitrogen contents in the milk produced from cows under HS. We slightly modified the statement and added some new statements to the revised manuscript. Please see the revised version of our manuscript (Line 281-286)
- Why are the findings on goat milk presented? There are plenty of literary sources in the field of cow's milk.
Response: Correct. The reference has now updated to a related article in cow. Please see the revision file for your approval (Line 301).
L.306 – 308 - his statement is not true, literary sources exist
Response: We're not sure if we fully grasp what the reviewer suggested in the comment above. However, using the discussion part of the reference we cited in the manuscript, we adjusted the statement as best as we could. In the published article (Ref. no 38, Humer et al., 2015), they mentioned that parity affected reticuloruminal temperature. They reported that it may be because older cows spent 1 h more per day with a reticuloruminal temperature >39.5°C, when compared with primiparous animals. The mechanism behind this phenomenon could be higher production in multiparous Holstein cows than that of primiparous animals result in higher activity and temperature of reticuloruminal of multiparous animals. Please see the revised version of our manuscript for your kind perusal (Line 326-330).
We agree with the statement that mentioned in the earlier study (Humer et al., 2015). In order to comply with the reviewer’s comment, we slightly modified the statement and added some new statement to the revised manuscript. Please see the revised version of our manuscript (Line 326-330) for your kind perusal.
Abstracts:
- L.18 – 19: no molecular mechanisms were pursued.
Response: Correct. We agree with the reviewer. We revised the sentence. Please see the revision file for your approval (Line 18-19).
- L.41 – 45: the conclusion is wrong - the negative impact on milk fat was not confirmed, unlike other studies.
Response: Yes, thank you very much for pointing this out. As the reviewer said, it is not that milk fat had a negative effect. We revised the sentence according to this comment. Please see the revision file for your approval (Line 41-45).
Although the authors obtained very valuable data, the manuscript processing needs to be revised and simpler goals should be set. Manuscripts can be on two topics due to the data obtained.
Once more, we want to express our gratitude to the reviewer for their insightful recommendations, which were all taken into account to improve the quality of our presentation.

Reviewer 2 Report
Review Animals
Evaluation of Heat Stress Effects in Different Geographical Areas on Milk and Rumen Characteristic in Holstein Dairy Cows Using Robot Milking and Rumen Sensors: A Survey in South Korea
General comments:
The manuscript studied the effects of an adjusted temperature-humidity index (THI) in different geographical areas on milk yield, fat and protein, rumen temperature, and activity in lactating Holstein cows using robotic milking and rumen sensors. The paper is well written and clear, and data are well explored in the results.
Particular comments
Lines 30-35: Could you please summarize your milk yield and quality findings? It appears confusing the way it is written. Either that or add this information (overall milk yield and quality) in the text.
Line 258-259: “It seems that the variation in milk yield per milking time is the most critical factor.” Please clarify this point since quality might also be important in the dairy industry.
Please find a few suggestions in the attached file.

Author Response
( ) I would not like to sign my review report
(x) I would like to sign my review report
English language and style
( ) Extensive editing of English language and style required
( ) Moderate English changes required
(x) English language and style are fine/minor spell check required
( ) I don't feel qualified to judge about the English language and style
Yes |
Can be improved |
Must be improved |
Not applicable |
|
Does the introduction provide sufficient background and include all relevant references? |
(x) |
( ) |
( ) |
( ) |
Are all the cited references relevant to the research? |
(x) |
( ) |
( ) |
( ) |
Is the research design appropriate? |
(x) |
( ) |
( ) |
( ) |
Are the methods adequately described? |
(x) |
( ) |
( ) |
( ) |
Are the results clearly presented? |
(x) |
( ) |
( ) |
( ) |
Are the conclusions supported by the results? |
(x) |
( ) |
( ) |
( ) |
Comments and Suggestions for Authors
Review Animals
Evaluation of Heat Stress Effects in Different Geographical Areas on Milk and Rumen Characteristic in Holstein Dairy Cows Using Robot Milking and Rumen Sensors: A Survey in South Korea
General comments:
The manuscript studied the effects of an adjusted temperature-humidity index (THI) in different geographical areas on milk yield, fat and protein, rumen temperature, and activity in lactating Holstein cows using robotic milking and rumen sensors. The paper is well written and clear, and data are well explored in the results.
Response: Thank you for your interest in our research.
Particular comments
Lines 30-35: Could you please summarize your milk yield and quality findings? It appears confusing the way it is written. Either that or add this information (overall milk yield and quality) in the text.
Response: We already written down the requested information in detail in the abstract (Line 28-35) and in the bottom of the abstract (Line 41-43). More detailed results cannot presented to the abstract at this stage due to the limitations of the words. Additionally, we wrote it down in the result section in full details (Line 177-209).
Line 258-259: “It seems that the variation in milk yield per milking time is the most critical factor.” Please clarify this point since quality might also be important in the dairy industry.
Response: Yes. Thank you very much for pointing this out. We modified the sentence for more clarifications. Please see the revision file for your approval (Line 270-271).
Once again, we are thankful to the reviewer for the insights comments that all were taken into consideration to improve the quality of our presentation.

Reviewer 3 Report
Title: Evaluation of Heat Stress Effects in Different Geographical Areas on Milk and Rumen Characteristic in Holstein Dairy Cows Using Robot Milking and Rumen Sensors: A Survey in South Korea
General Comments:
The aim of study is to use the robot milking and rumen sensors to evaluate the effect of heat stress on dairy cows in the South Korea. It is meaningful to evaluate the physiological state of dairy cows using the advanced production equipment from the view of time and space. But the authors must improve several points:
1. I think it is unreasonable for the author to only focus on the heat stress. Because the study about heat stress needs the strict control group, including the same lactating period and diet. In my opinion, the aim of the study can be modified as “A time and space survey of lactating performance and rumen function using robot milking and rumen sensors”. The heat stress could be a potential factor affecting the physiological state of dairy cows.
2. The details of feeding and management need to be provided.
3. It may be better to display the table in the form of line chart.
Specific comments:
Line80-81: I can't understand this sentence and its innovation. Do “the effect of Robot Milking and Rumen Sensors” have any contact with “heat stress”?
Line95: Are the TMR of the 4 farms the same?
Line97: Please providing the details of feeding and management for the 4 farms, including milking frequency, semi enclosed or enclosed sheds, etc.
Line106-107: How to define the “activity data”.
Line143-144: How about the Lactating period?
Line147: Is it better to use line chart for time data?
Line226: data sources.
Line238-239: How about the enclosed sheds?
Line292-293: Please provide references.
Line341-342: Lactation depends on rumen microbial activity. Could you provide ref. to support your opinion.
Other comments:
Introduction
Could you provide the differences between South Korea and other regions to highlight the advantages of the study.
Materials and Methods
Please provide more detailed information for dairy cows and 4 farms of the study.
Please provide the calculation formula for 3.5% FCM and ECM.
Discussion
Please discuss lactating period, because your study could not avoid the effect of lactating period.
Author Response
Open Review
( ) I would not like to sign my review report
(x) I would like to sign my review report
English language and style
( ) Extensive editing of English language and style required
( ) Moderate English changes required
( ) English language and style are fine/minor spell check required
(x) I don't feel qualified to judge about the English language and style
Yes |
Can be improved |
Must be improved |
Not applicable |
|
Does the introduction provide sufficient background and include all relevant references? |
( ) |
(x) |
( ) |
( ) |
Are all the cited references relevant to the research? |
(x) |
( ) |
( ) |
( ) |
Is the research design appropriate? |
( ) |
(x) |
( ) |
( ) |
Are the methods adequately described? |
( ) |
(x) |
( ) |
( ) |
Are the results clearly presented? |
( ) |
(x) |
( ) |
( ) |
Are the conclusions supported by the results? |
( ) |
(x) |
( ) |
( ) |
Comments and Suggestions for Authors
Title: Evaluation of Heat Stress Effects in Different Geographical Areas on Milk and Rumen Characteristic in Holstein Dairy Cows Using Robot Milking and Rumen Sensors: A Survey in South Korea
General Comments:
The aim of study is to use the robot milking and rumen sensors to evaluate the effect of heat stress on dairy cows in the South Korea. It is meaningful to evaluate the physiological state of dairy cows using the advanced production equipment from the view of time and space. But the authors must improve several points:
- I think it is unreasonable for the author to only focus on the heat stress. Because the study about heat stress needs the strict control group, including the same lactating period and diet. In my opinion, the aim of the study can be modified as “A time and space survey of lactating performance and rumen function using robot milking and rumen sensors”. The heat stress could be a potential factor affecting the physiological state of dairy cows.
Response: Thank you for your interest in our research. This is a survey study, as the reviewer has already stated. Unlike research articles that must have a strict control group, in survey studies, it is almost impossible to provide a strict control group when using data from many farms in various geographic locations. The time of the survey is already mentioned in the original manuscript (see Table 1 for your kind information). The data of heat stress were used from the months of June to August.
- The details of feeding and management need to be provided.
Response: We now added clarification sentences into the revised manuscript regarding the TMR compositions and housing system in the farms. Since the farms were in different areas and the exact compositions of TMRs are private information belonging to each farm, we don’t have detailed information. However, we asked farmers to provide us with some data to clarify the general conditions of the housing system and also the TMR compositions. We understand that the TMR compositions including forage to concentrate ratio in all farms were similar (F:C ratio of 60:40%), with CP:14~15 (Line 103-106). The housing system were the same as we have chosen the farm that had similar conditions in regard with housing (free-stall housing with open barn having ceiling to use for shade or protecting from rains) (Line 106-108). Please see the revision file for your approval.
- It may be better to display the table in the form of line chart.
Response: Thank you for your good suggestion. In our opinion, each farm has a different temperature, humidity and THI, so we think it would be less confusing to present it as a table rather than a line chart.
Specific comments:
Line80-81: I can't understand this sentence and its innovation. Do “the effect of Robot Milking and Rumen Sensors” have any contact with “heat stress”?
Response: Yes, thank you for pointing this out. We revised the sentence according to the given comment. Please see the revision file for your approval (Line 86-88).
Line95: Are the TMR of the 4 farms the same?
Response: Since the farms were in different areas and the exact compositions of TMRs are private information belonging to each farm, we don’t have detailed information. However, we asked farmers to provide us with some data to clarify the general conditions of the housing system and also the TMR compositions. We understand that the TMR compositions including forage to concentrate ratio in all farms were similar (F:C ratio of 60:40%), with CP:14~15 (Line 103-106). Please see the revision file for your approval.
Line97: Please providing the details of feeding and management for the 4 farms, including milking frequency, semi enclosed or enclosed sheds, etc.
Response: All dairy cows in the farms were milked twice a day (Line 113) and the farms were composed of semi-enclosed sheds. In addition, we now added clarification sentences into the revised manuscript regarding the TMR compositions and housing system in the farms. Since the farms were in different areas and the exact compositions of TMRs are private information belonging to each farm, we don’t have detailed information. However, we asked farmers to provide us with some data to clarify the general conditions of the housing system and also the TMR compositions. We understand that the TMR compositions including forage to concentrate ratio in all farms were similar (F:C ratio of 60:40%), with CP:14~15 (Line 103-106). The housing system were the same as we have chosen the farm that had similar conditions in regard with housing (free-stall housing with open barn having ceiling to use for shade or protecting from rains) (Line 106-108).
Line106-107: How to define the “activity data”.
Response: Regarding the Instruction for checking activity data, we have already clarified this issue. (Line 142-156). Please see the revision file for your approval.
Line143-144: How about the Lactating period?
Response: We chose the entire lactation on each farm, but to prevent the results from being confounded, we kept the number of cows subjected to data collection in each lactation phase (early lactating, mid-lactating, and late lactating) comparable.
Line147: Is it better to use line chart for time data?
Response: Thank you for you’re the good suggestion. In our opinion, each farm has a different temperature, humidity, and THI, so we think it would be less confusing to present it as a table rather than a line chart.
Line226: data sources.
Response: Thank you for point this out. We don`t need this sentence, so we deleted it. Please see the revision file for your approval
Line238-239: How about the enclosed sheds?
Response: The temperature and humidity were remained the same during the night even in the enclosed sheds.
Line292-293: Please provide references.
Response: Correct. We added a reference in the sentence. Please see the revision file for your approval (Line 308-309).
Line341-342: Lactation depends on rumen microbial activity. Could you provide ref. to support your opinion.
Response: Since the number of cows in each lactation phase in this survey were similar (early, mid, and late lactation phase), we could assume that the average microbial activity in all cows of all farms were similar or if little variation existed does have minor effect on the obtained results (Line 364-365).
Heat stress has both direct and indirect impacts on the health and welfare of animals by impairing the rumen and intestinal mechanisms, thereby reducing the efficiency of feed utilization. Regulating rumen microbial fermentation is essential for the use of nutrients and additives to alleviate the negative effects of heat stress.
Source: Kim et al., (28 February 2022). “Heat stress: Effects on Rumen Microbes and Host Physiology, and Strategies to Alleviate the Negative Impacts on Lactating Dairy Cows”. frontiers in Microbiology. 13. 804562. Doi:10.3389/fmicb.2022.804562.
Other comments:
Introduction
Could you provide the differences between South Korea and other regions to highlight the advantages of the study.
Response: Thank you for your interest in our research In contrast to other regions, South Korea is surrounded by water, thus the temperature and humidity were high. Given the aforementioned information, South Korea has different environmental circumstances than other countries, making the country's temperature and humidity essential while researching heat stress in Holstein cows. We add to sentence in manuscript (Line 76-80). Please see the revision file for your approval.
Materials and Methods
Please provide more detailed information for dairy cows and 4 farms of the study.
Response: The housing system were the same as we have chosen the farm that had similar conditions in regard with housing (free-stall housing with open barn having ceiling to use for shade or protecting from rains) (Line 103-108). Please see the revision manuscript for your kindly approval.
Please provide the calculation formula for 3.5% FCM and ECM.
Response: Yes, we agree with the reviewer comment. As the reviewer said, the equation by calculating 3.5% FCM and ECM were written in the materials and methods (Line 139-142). Please see the revision manuscript for your approval.
Discussion
Please discuss lactating period because your study could not avoid the effect of lactating period.
Response: We chose the entire lactation on each farm, but to prevent the results from being confounded, we kept the number of cows subjected to data collection in each lactation phase (early lactating, mid-lactating, and late lactating) comparable.
Once more, we want to express our gratitude to the reviewer for their insightful recommendations, which were all taken into account to raise the caliber of our presentation.

Reviewer 4 Report
Why do the authors associate the Robot Milking System with Rumen Sensors? It is difficult to understand.
You had to change the title because it did not correspond to the study's content. This did not appear to be a survey study.
This work appeared to be divided into two sections, one looking for the effect of THI on milk production and composition and the other looking for rumen sensors on rumen characteristics.
The authors must provide more data, such as DMI, in the first set about THI in order to determine why milk production decreases as THI increases.
Before analyzing the correlation, the authors must provide more data on milk production and milk composition in the second set.
The authors should provide rumen characteristics derived from rumen sensors.
Please recheck the data from farm D in table 3.
Please explain why the authors analyzed Pearson's correlation coefficient of Farm A and B separately.
Author Response
Comments and Suggestions for Authors
Why do the authors associate the Robot Milking System with Rumen Sensors? It is difficult to understand.
Response: We understand the reviewer’s point of view. To address this, we should note that the rumen activity has a significant role in milk production and characteristics. Thus, study of rumen activity using rumen sensor (in smart farming systems) seems crucial to address the effect of rumen activity on milk production and its characteristics, particularly during heat stress during which the cows’s core body temperature is higher than the normal conditions. This high body temperature may confound the optimum health and activity of microbiota in the rumen and consequently may have negative effects on production performance (herein milk yield and compositions). On the other hand, robot milking system has received a lot of attention recently due to its effect on the cows’ choice regarding milking frequency. The frequency of milking has a considerable impact on the amount of milk produced and its subsequent composition in dairy cows, as the reviewer may better comprehend and as addressed in several publications. Due to the decrease in milk supply, these effects are especially significant when investigating under heat stress. Given the above reasons, we thought that the combination of using robot milking system and rumen sensors is pivotal to study together since both of these factors are considered a great reasons behind milk production, health and milk compositions in dairy cows and most importantly during heat stress conditions.
You had to change the title because it did not correspond to the study's content. This did not appear to be a survey study.
Response: Regarding the reviewer's concern with the title and its representation of the study's substance, we respectfully ask to keep the present title or make minor changes in the second round of editing if the reviewer point out the proper modifications. To our understanding, this is a survey study and there are other studies that, when combined, provide a study's flow by supplying data from numerous farms and animals. . So, it is not a research paper but the survey. However, we are open to further modification in the next round of revision if the respected reviewer kindly suggests us some more appropriate title.
This work appeared to be divided into two sections, one looking for the effect of THI on milk production and composition and the other looking for rumen sensors on rumen characteristics.
Response: Our goal was to establish a single study flow. Therefore, although our presentation looks to do so, we did not attempt to split the study. Thus, we are not sure what exactly the reviewer asks us to do. This is a study to confirm milk and rumen performance according to the THI environment in summer. In addition, it is to confirm whether there is a difference depending on the parity difference.
The authors must provide more data, such as DMI, in the first set about THI in order to determine why milk production decreases as THI increases.
Response: Since this is a survey paper, each farm's Holstein cows in Korea received an average of roughly 30 kg of feed, however it was not recorded on a daily basis.
Before analyzing the correlation, the authors must provide more data on milk production and milk composition in the second set.
Response: The data related to milk production and compositions have already been presented in table 3.
The authors should provide rumen characteristics derived from rumen sensors.
Response: Thank you for your interest our research. We added sentences about rumen characteristics derived from rumen sensors. Please see the revision file for your approval. (Line 310-313)
Please recheck the data from farm D in table 3.
Response: The farm D in Table 3 did not include some missing data from other seasons except summer.
Please explain why the authors analyzed Pearson's correlation coefficient of Farm A and B separately.
Response: We agree with the reviewer regarding this comment. Thank you for this insightful comment. However, we did this separate analysis on purpose. The reason why is that, although the environmental, feeding, and housing management in the farms were similar if not the same. Using separate correlation analysis, we aimed to see whether there are other unknown influential factors such as farm locations, wind speed, genotype etc. affecting the results. The comparable pattern in the collected results supports the similarity of the cows in the farms, as the reviewer can clearly observe from Tables 7 and 8 implying that the other unknown influential factors could not significantly affect the output of this study.
Once more, we want to express our gratitude to the reviewer for their insightful recommendations, which were all taken into account to raise the caliber of our presentation.

Round 2
Reviewer 1 Report
The authors corrected the article based on my comments. The manuscript can be published.
Author Response
Open Review
( ) I would not like to sign my review report
(x) I would like to sign my review report
English language and style
( ) Extensive editing of English language and style required
( ) Moderate English changes required
( ) English language and style are fine/minor spell check required
(x) I don't feel qualified to judge about the English language and style
Yes |
Can be improved |
Must be improved |
Not applicable |
|
Does the introduction provide sufficient background and include all relevant references? |
(x) |
( ) |
( ) |
( ) |
Are all the cited references relevant to the research? |
(x) |
( ) |
( ) |
( ) |
Is the research design appropriate? |
(x) |
( ) |
( ) |
( ) |
Are the methods adequately described? |
(x) |
( ) |
( ) |
( ) |
Are the results clearly presented? |
(x) |
( ) |
( ) |
( ) |
Are the conclusions supported by the results? |
(x) |
( ) |
( ) |
( ) |
Comments and Suggestions for Authors
The authors corrected the article based on my comments. The manuscript can be published.
Response: Once again, thank you so much for accepting our work for publication in the present form.
Once more, we want to express our gratitude to the reviewer for their insightful recommendations, which were all considered to improve the quality of our presentation.

Reviewer 3 Report
lines 116-117: Provide the developer of "software program". I still think “rumen activity” is abstract, “microbial number”? or “enzyme activity” or others?
lines 182-183: The table2 need to be improved
lines 239: What is the data ("3.2% and 4.1%, respectively") source
Author Response
Open Review
( ) I would not like to sign my review report
(x) I would like to sign my review report
English language and style
( ) Extensive editing of English language and style required
( ) Moderate English changes required
( ) English language and style are fine/minor spell check required
(x) I don't feel qualified to judge about the English language and style
Yes |
Can be improved |
Must be improved |
Not applicable |
|
Does the introduction provide sufficient background and include all relevant references? |
(x) |
( ) |
( ) |
( ) |
Are all the cited references relevant to the research? |
(x) |
( ) |
( ) |
( ) |
Is the research design appropriate? |
(x) |
( ) |
( ) |
( ) |
Are the methods adequately described? |
( ) |
(x) |
( ) |
( ) |
Are the results clearly presented? |
( ) |
(x) |
( ) |
( ) |
Are the conclusions supported by the results? |
(x) |
( ) |
( ) |
( ) |
Comments and Suggestions for Authors
lines 116-117: Provide the developer of "software program". I still think “rumen activity” is abstract, “microbial number”? or “enzyme activity” or others?
Response: Thank you again for your comment. We added the company name that developed the software program. Please see the revision file for your approval (Line 117). Rumen activity was used to measure changes in dairy cows' rumen movement.
lines 182-183: The table2 need to be improved
Response: Correct. Thank you for your point. We revised the format of the table 2. Please see the revision file for your approval.
lines 239: What is the data ("3.2% and 4.1%, respectively") source
Response: As shown in below paper (cited number 1), the largest cluster was `Animal` (49%). The results from the analysis for cluster `Animal` showed the high relevance of topics related to the animal health and welfare. We added the citation at the end of the sentence to avoid the readers’ confusion and to refer the avid readers of Animals to the cited reference.
Source: Cogato, A.; Brščić, M.; Guo, H.; Marinello, F.; Pezzuolo, A. Challenges and tendencies of automatic milking systems (AMS): A 20-years systematic review of literature and patents. Animals 2021, 11, 356.
We would like to thank the reviewer once more for his or her keen eyes and comments, which were extremely valuable and could improve the quality of our presentation.

Reviewer 4 Report
It was still unclear how robot milking related to rumen sensors that reflect milk production and compositions.
Why did you only have rumen sensor data from July to September? What about other months? Is there a difference in rumen temperature and activity in the other months?
Can you describe that rumen sensor data related to milk yield and compositions based on Pearson's correlation coefficients of considered traits for daily milking and rumen characteristics of farms A and B?
You stated that when ruminal activity is reduced, the rumen produces less volatile fatty acids [50]. Do you have a rumen activity score that affects VFA production? Line 337-338.
Please recheck the milk protein in the first parity. (Table 4)
Author Response
Comments and Suggestions for Authors
It was still unclear how robot milking related to rumen sensors that reflect milk production and compositions.
Response: We missed what the respected reviewer asked us to explain in the first part of the comment (how robot milking related to rumen sensors). We did, however, respond to the second part of the comment (reflection of rumen sensors for measuring rumen activity and its relationship with milk production and compositions). If we understand the first part of the comment correctly, the reviewer inquires about the relationship between robot milking and rumen sensor. To address this, we should clarify that we did not claim a link between rumen sensors and robot milking. What we claimed is their relationship with milk production and composition, which is discussed further below:
Rumen activity is thought to play an important and indirect role in milk production and characteristics. Given the close relationship between rumen activity and rumen digesta, microbial populations, and rumen health (rumen temperature, optimal populations of rumen microorganisms, etc.), any changes or negative effect on rumen activity can lead to a decrease in feed intake, which has a negative effect on milk yield and characteristics (cited number 3). Thus, studying rumen activity using rumen sensors is critical, as data on the relationship between rumen activity and milk yield and compositions is limited, which is one of the study's objectives. This study is highlighted in particular because it focuses on heat stress, which occurs when animals are exposed to high body temperatures and have less rumen activity, resulting in a reduced ability to dissipate heat from the body and, as a result, a decrease in feed intake. High body temperature may interfere with the optimal health and activity of microbiota in the rumen, resulting in poor production performance (herein milk yield and compositions). The frequency of milking, on the other hand, has a significant impact on the amount of milk produced and its subsequent composition in dairy cows. These effects are especially significant when investigating under heat stress due to the decrease in milk supply. Given the above reasons, we believe that the combination of using a robot milking system and rumen sensors is critical to study together because both of these factors are important in milk production, health, and milk compositions in dairy cows, particularly during heat stress conditions. .
Why did you only have rumen sensor data from July to September? What about other months? Is there a difference in rumen temperature and activity in the other months?
Response: There is no data from January to June. Because the rumen sensor was implanted in July. In addition, the goal of this study was to investigate rumen sensor data during heat stress, which occurs in Korea during the summer (July-September). In the future, we will conduct additional research to compare the data by season.
Can you describe that rumen sensor data related to milk yield and compositions based on Pearson's correlation coefficients of considered traits for daily milking and rumen characteristics of farms A and B?
Response: The Pearson correlation measures the strength of the linear relationship between two variables. In this regard, we aimed to find out the correlation between these two variables (herein rumen sensors and milk yield) using Pearson's correlation coefficients. There are other types of correlations analyses such as regression models (which is not applicable here). In statistics we use correlation to denote association between two quantitative variables. A correlation coefficient measures the degree of association. . It is sometimes called Pearson’s correlation coefficient after its originator and is a measure of linear association.
The correlation coefficient is measured on a scale that varies from + 1 through 0 to – 1. Complete correlation between two variables is expressed by either + 1 or -1. When one variable increases as the other increases the correlation is positive; when one decreases as the other increases it is negative. Complete absence of correlation is represented by 0.
Given the above information, in order to determine the relationship between rumen sensor and milk yield (as we previously explained the scientific relation between these two factors), we have an option to quote this correlation using the Pearson's correlation coefficients.
You stated that when ruminal activity is reduced, the rumen produces less volatile fatty acids [50]. Do you have a rumen activity score that affects VFA production? Line 337-338.
Response: There is currently no data on the rumen activity score and its relationship with volatile fatty acid production. This was an intriguing comment that could serve as a platform for further investigation.
Please recheck the milk protein in the first parity. (Table 4)
Response: Thank you for your point. We re-checked the raw data again and revised it to the data. Because it is a primiparous cows, The milk protein content of primiparous cows may be slightly higher than that of multiparous cows.
We would like to thank the reviewer once more for his or her keen eyes and comments, which were extremely valuable and could improve the quality of our presentation. We hope that with this final amendment, we could satisfy the reviewer points of view and that of our manuscript can be accepted for publication.
